# The Role of Pigments and Cryptochrome 1 in the Adaptation of *Solanum lycopersicum* Photosynthetic Apparatus to High-Intensity Blue Light

**DOI:** 10.3390/antiox13050605

**Published:** 2024-05-15

**Authors:** Aleksandr Ashikhmin, Pavel Pashkovskiy, Anatoliy Kosobryukhov, Alexandra Khudyakova, Anna Abramova, Mikhail Vereshchagin, Maksim Bolshakov, Vladimir Kreslavski

**Affiliations:** 1Institute of Basic Biological Problems, Russian Academy of Sciences, Institutskaya Street 2, Pushchino 142290, Russia; ashikhminaa@gmail.com (A.A.); kosobr@rambler.ru (A.K.); s_t_i_m_a_@mail.ru (A.K.); lfbv22@gmail.com (M.B.); 2K.A. Timiryazev Institute of Plant Physiology, Russian Academy of Sciences, Botanicheskaya Street 35, Moscow 127276, Russia; pashkovskiy.pavel@gmail.com (P.P.); ann.kiedis2000@gmail.com (A.A.); mhlvrh@mail.ru (M.V.)

**Keywords:** cryptochrome 1, high irradiance, blue light, photomorphogenetic mutants, *Solanum lycopersicum*, photosynthetic apparatus, pigments

## Abstract

The effects of high-intensity blue light (HIBL, 500/1000 µmol m^−2^s^−1^, 450 nm) on *Solanum lycopersicum* mutants with high pigment (*hp*) and low pigment (*lp)* levels and cryptochrome 1 (cry1) deficiency on photosynthesis, chlorophylls, phenols, anthocyanins, nonenzymatic antioxidant activity, carotenoid composition, and the expression of light-dependent genes were investigated. The plants, grown under white light for 42 days, were exposed to HIBL for 72 h. The *hp* mutant quickly adapted to 500 µmol m^−2^s^−1^ HIBL, exhibiting enhanced photosynthesis, increased anthocyanin and carotenoids (beta-carotene, zeaxanthin), and increased expression of key genes involved in pigment biosynthesis (*PSY1*, *PAL1*, *CHS*, *ANS*) and PSII proteins along with an increase in nonenzymatic antioxidant activity. At 1000 µmol m^−2^s^−1^ HIBL, the *lp* mutant showed the highest photosynthetic activity, enhanced expression of genes associated with PSII external proteins (*psbO*, *psbP*, *psbQ*), and increased in neoxanthin content. This mutant demonstrated greater resistance at the higher HIBL, demonstrating increased stomatal conductance and photosynthesis rate. The *cry1* mutant exhibited the highest non-photochemical quenching (NPQ) but had the lowest pigment contents and decreased photosynthetic rate and PSII activity, highlighting the critical role of CRY1 in adaptation to HIBL. The *hp* and *lp* mutants use distinct adaptation strategies, which are significantly hindered by the *cry1* mutation. The pigment content appears to be crucial for adaptation at moderate HIBL doses, while CRY1 content and stomatal activity become more critical at higher doses.

## 1. Introduction

Light is one of the most significant factors affecting photosynthesis and phytochemical synthesis, and in many plant species, it leads to the accumulation of different plant metabolites, particularly leaf pigments [1,2]. Moreover, light quality is one of the key variables influencing the content of different pigments [3,4,5,6]. Among the different spectral regions, blue light (BL) is involved in many processes, such as phototropism, photomorphogenesis, stomatal opening, and leaf photosynthesis [6,7].

For example, the application of supplemental high-intensity blue light (HIBL) enhanced the contents of phenolic compounds, flavonoids in the leaves of *Hedyotis corymbosa* (L.) [8], and carotenoids such as violaxanthin and zeaxanthin involved in the process of dissipating absorbed energy into heat in *Lactuca sativa* [9]. Additionally, BL induces the synthesis of important pigments, Chls and β-carotene [5]. In addition, BL can enhance photosynthesis.

Thus, the study by Le et al. (2021) showed that BL enhanced photosynthesis, likely by increasing stomatal conductance and the rate of photosynthetic electron transport [8]. Additionally, BL improved protection by decreasing the quantum yield of nonphotochemical loss. Wang et al., 2015 showed that BL plays a key role in the development of the photosynthetic apparatus (PA) in *Cucumis sativus* plants grown under low light conditions [4].

The initial inactive form of phytochrome (P_r_) and physiologically active form of phytochrome (P_fr_) absorb light not only in the red light (RL) region but also in the BL region. Hence, phytochrome (PHY) effects can be observed under BL [7]. PHY is able to regulate photosynthesis, chloroplast formation, and the accumulation of different metabolites, including leaf pigments [10,11].

The accumulation of leaf pigments primarily depends on the state of photoreceptors such as cryptochromes (CRYs) and PHYs [12]. Thus, a noticeably lower content of carotenoids and UV-absorbing pigments was detected in mutant *Arabidopsis thaliana* plants than in wild type (WT) plants grown on HIBL supplemented with CRY1 deficiency, and a decrease in the activity of a number of antioxidant enzymes that protect the plant from oxidative stress was also detected [13]. The important role of CRY1 during the action of high-intensity light (HIL) on the photosynthetic processes of *A. thaliana* plants has also been shown [14]. The authors suggest a novel function of CRY1 in plant responses to HIL and in inducing photoprotective mechanisms.

As light intensity increases, elevated levels of reactive oxygen species (ROS) are observed. This not only leads to damage but also triggers ROS signaling [15,16,17]. ROS scavenging mechanisms play an important role in neutralizing the negative effects of ROS on the PA. Some studies showed that the degree of PSII photodamage is caused by incident light intensity but remains unaffected by various environmental conditions, which affect PSII recovery by inhibiting D1 protein biosynthesis [18]. To defend against oxidative stress, plants have developed a complex system of low-molecular-weight antioxidants and antioxidant enzymes. Carotenoids and phenolic compounds, including anthocyanins, can serve not only as antioxidants, which can inactivate free radicals and protect cells from ROS, but also as optical filters that absorb excess light [19,20,21].

Zulfiqar et al. (2021) reported that carotenoids absorb light to transfer it to photosynthetic reaction centers and protect PA from a damage induced by high light [22]. Particularly important for the protection of PA are xanthophylls such as lutein and zeaxanthin, as well as a pigment from the carotene class, β-carotene. Zeaxanthin is formed in the light in the xanthophyll cycle and plays a major role in nonphotochemical quenching in the antenna of PSII [23]. β-Carotene can neutralize the triplet state of Chl, thus protecting PA from the formation of ROS [24,25]. Lutein is involved in the stabilization of antenna proteins and quenches the triplet state of Chl, and there are also reports of its possible participation in the NPQ mechanism [23].

Carotenoids and anthocyanins are known to have adaptive functions in *S. lycopersicum* plants deficient in pigments under high irradiance, as described in the study of Ashikhmin et al. (2023) [26]. In this case, a high-pigment mutant (*hp*) demonstrated elevated contents of photosynthetic pigments and anthocyanins, whereas a low-pigment mutant *(lp*) demonstrated an elevated content of ultraviolet-absorbing pigments but a decreased content of photosynthetic pigments and anthocyanins. Compared with the WT and *lp* mutants, the *hp* mutants were more resistant to high white light (2000 μmol (photons) m^−2^s^−1^, 72 h). Additionally, the *hp* mutant showed an increase in leaf thickness and water content under high irradiance, which suggests high adaptability to stress and reduced photoinhibition.

CRY plays a key role in plant growth and metabolic and photomorphogenetic processes, including the movement of chloroplasts, flowering, the regulation of stomatal activity, seed germination, deetiolation, the inhibition of stem growth, and the synthesis of photosynthetic pigments, flavonoids, and anthocyanins [27,28]. It has been suggested that photoreceptors take part in the induction of the accumulation of various pigments and thus help plants and their PAs adapt to stress conditions, primarily intense white light [26]. Moreover, only CRY1, not CRY2, is able to protect against HIL, but the role of CRY1 in these studies was insufficiently examined. Therefore, it would be interesting to understand the role of pigment levels under conditions in which the content of active CRY1 is the most important, that is, under high blue irradiance. The important role of CRY1 in plants and its PA response to high irradiance has been described previously [13,14,26]. However, many details are unclear.

This study investigated the role of CRY1 in plants grown under HIBL, where in our opinion, CRY1 is the most important since this photoreceptor has the highest activity. One of the key mechanisms underlying the participation of CRY1 in different processes could be the enhanced expression of light-activated genes. In addition, studies of the specific mechanisms that help to adapt to HIL by using different strategies to reduce oxidative stress induced by high light may be promising. Therefore, some model systems, such as plants under BL conditions and mutants, could be useful for identifying the roles of various pigments and low-molecular-weight antioxidants under conditions of photoinhibition induced by light.

Tomato (*S. lycopersicum* L.) is an important crop species in which the molecular functions and roles of photoreceptors in different processes have been sufficiently studied. Tomato plants have the following five PHYs: phyA, phyB1, phyB2, phyE, and phyF [29]. Additionally, tomato plants contain two main CRYs: CRY1A and CRY2 [28]. Moreover, CRYs in tomato strongly affect seed and seedling development, especially hypocotyl elongation, root growth, and development and the accumulation of photosynthetic pigments [28].

Notably, the elevated pigment content indicated in tomato *hp*1 and *hp*2 mutants led to increased photosynthesis, decreased stomatal limitation, and increased carboxylation rates [30]. These mutants are deficient in the DDB1 (*hp1*) and DET1 (*hp2*) genes, which regulate photomorphogenesis and other plant functions, particularly photosynthesis [26]. However, a reduction in the number of flowers and fruits and a delay at the beginning of flowering were observed in such plants. DET1 is a protein that is a negative regulator of photomorphogenesis, and DET1 mutation leads to enhanced photomorphogenesis and elevated accumulation of pigments [31]. In contrast, in the case of a low (with abscisic acid insensitive (*ABI3*) mutation) pigment mutant [26], reduced pigment savings can be expected.

We suggest that plant susceptibility to HIL exposure depends on the activity and content of photoreceptors such as CRY1 and the content of different pigments located in leaves. Therefore, the aim of the present work was to explore the role of leaf pigments and CRY1 in the adaptation of the tomato (*S. lycopersicum* L.) PA to different doses of BL. Using *cry1*, high-pigment (*hp*, with DET1 mutation), and low-pigment (with *ABI3* mutation) tomato mutants, we evaluated the relationship between the content of pigments (carotenoids and their composition and flavonoids, including anthocyanins) and PA resistance to HIBL. We also suggest that high pigment and *cry1* active form contents and maintenance of stomatal activity are critical for tomato adaptation to high BL.

## 2. Materials and Methods

### 2.1. Plant Materials and Experimental Design

Wild-type *Solanum lycopersicum* L. plants (Moneymaker cultivar, LA2706) and photomorphogenic high-pigment (*hp*, LA3005, mutation of the De-etiolated 1 (DET1) gene), low-pigment (*lp*, LA3617 mutation of the Abscisic-acid-insensitive 3, *ABI3* gene) and cryptochrome deficiency (*cry1*, LA4359) mutants were used in the experiment. Seeds were obtained from the Tomato Genetics Resource Center (TGRC) (University of California, Davis, CA, USA). The plants were grown for 42 days in a thermostatically controlled chamber with a 12 h photoperiod at a temperature of 25 ± 1 °C during the day and 21 ± 1 °C during the night. Then, we used the cuttings and cloned the plants, which were subsequently grown for 10 days under weak room light for rooting. Then, some of the plants were grown under white fluorescent lamps (WFL) (Philips, Pila, Poland) for 42 days in 8 cm × 8 cm × 10 cm vessels filled with perlite under a 12 h photoperiod and a light intensity of 250 µmol (photons) m^−2^s^−1^ under the above temperature conditions. Four plants of each variant were planted in each vessel. During the cultivation period, the plants were watered with half-strength Hoagland solution. After growth, the plants were continuously irradiated with high-intensity blue light for 72 h using LEDs (I = 500 ± 20 and 1000 ± 50 μmol (photons) m^−2^s^−1^) (λ_max_ = 457 nm, half-width = 26 nm). The spectral characteristics of the light sources (Figure 1) were determined using an AvaSpecULS2048CL-EVO spectrometer (Avantes, Apeldoorn, The Netherlands). At the end of the experiment, young leaves that formed under appropriate light conditions were selected for analysis and fixed in liquid nitrogen. Leaves of original plants grown under fluorescent lamps were used as controls. The majority of the analyses were conducted after 24 h, 48 h, and 72 h of HIBL treatment, and some of the analyses were conducted only after 72 h. Five to ten of the most developed leaves from the third and second tiers were used for the analysis. During the experiments, the photosynthetic parameters, G_s_, E, and pigment contents were determined.

### 2.2. Photosynthesis and PAM Measurements

The photosynthesis (P_n_) and transpiration (E) rates and stomatal conductance (G_S_) of the tomato leaves were determined using a LCPro+ portable infrared gas analyzer from ADC BioScientific Ltd. (Hoddesdon, UK), which was connected to a leaf chamber with an area of 6.25 cm^2^. The measurements were carried out at a saturating light intensity of 1000 μmol (photons) m^−2^s^−1^.

Fluorescence induction curves were obtained using a mini-PAM II fluorometer (Walz, Effeltrich, Germany) on plants dark-adapted for 30 min. Measurements involved a series of light exposures: measuring light at 0.5 μmol (photons) m^−2^s^−1^, actinic light at 190 μmol (photons) m^−2^s^−1^, and saturating pulses peaking at 474 nm with an intensity of 3000 μmol (photons) m^−2^s^−1^ and a duration of 800 ms, all of which used blue LEDs that also peaked at 474 nm. Parameter calculations based on chlorophyll fluorescence data were performed using WinControl-3 v3.32 software (Walz, Effeltrich, Germany). The values for F_0_, F_v_, F_m_, F_m′_, and F_0′_ were determined, as well as the PSII maximum quantum yield (F_v_/F_m_) and the effective quantum yield of PSII photochemistry Y(II) ((F_m′_ − F_t_)/F_m′_) and non-photochemical quenching NPQ ((F_m_ − F_m′_)/F_m′_) [32]. Here, F_m_ and F_m′_ are the maximum levels of chlorophyll fluorescence under dark and illuminated conditions, respectively. F_v_ is the photoinduced change in fluorescence, F_t_ is the level of steady-state chlorophyll fluorescence, and F_0_ is the initial level of chlorophyll fluorescence. In addition, Y(NO) and Y(NPQ) are the quantum yields of unregulated and regulated non-photochemical energy dissipation in PSII, respectively.

### 2.3. Contents of Photosynthetic Pigments, Phenols and Anthocyanins

The contents of chlorophyll *a* (Chl *a*), *b* (Chl *b*), and carotenoids (Car) were determined by a previously described method [33]. The samples were ground with 80% acetone in the dark. After centrifugation of the samples, the optical density of the solutions was measured using a Genesys 10 UV ultraviolet-visible spectrophotometer (VIS) (Thermo Fisher Scientific, Waltham, MA, USA) at wavelengths of 470, 646, and 663 nm. The content of photosynthetic pigments was determined using the following formula: Chl *a* = 12.25 × A663 − 2.79 × A646; Chl *b* = 21.50 × A646 − 5.10 × A663; and Car = (1000 × A470 − 1.82 × Chl *a* − 85.02 × Chl *b*)/198.

The anthocyanins were extracted and determined spectrophotometrically [34]. We used 0.10–0.15 g of leaf mass per sample, crushed it in liquid nitrogen, and incubated it in 600 μL of extraction buffer (methanol with the addition of 1% HCl) in an ultrasonic bath for 15 min and overnight at 4 °C in the dark.

The total phenolic content was determined spectrophotometrically using Folin and Ciocalteu’s phenol reagent (Sigma-Aldrich, Burlington, MA, USA, MDL number MFCD00132625) according to a previously described method [35]. The total phenolic content was expressed in gallic acid equivalents (GAE)—mg g^−1^ (FM or DM).

### 2.4. Trolox Equivalent Antioxidant Capacity

The Trolox equivalent antioxidant capacity (TEAC) was evaluated using a Hitachi-557 spectrophotometer (Kyoto, Japan) according to a previously described method [36] involving the reaction of methanolic extracts with 2,2′-azino-bis[3-ethylbenzothiazoline-6-sulfonic acid] diammonium salt (ABTS) (Sigma-Aldrich, Burlington, MA, USA, CAS No. 30931-67-0). The TEAC was expressed as μmol (Trolox) g^−1^ (FM).

### 2.5. HPLC Analysis of Carotenoids

Carotenoid analysis was performed on an HPLC device (Shimadzu, Kyoto, Japan) consisting of an LC-10ADVP pump with an FCV-10ALVP module, an SPD-M20A detector and a CTO-20 AC thermostat [26,37]. The separation of carotenoids was performed on a 4.6 mm × 250 mm reversed-phase column (Agilent Zorbax SB-C18, Agilent, Santa Clara, CA, USA) at 22 °C. The column was balanced with an acetonitrile/water mixture (90/10%, *v*/*v*). Then, the acetonitrile/water/ethyl acetate mixture (69.3%:7.7%:23%, *v*/*v*) was passed through the column in the first 3 min. The mixture was linearly substituted with pure ethyl acetate (37 min), and ethyl acetate was passed through the column for the next 3 min. The solvent feed rate was 1.0 mL/min. The carotenoids were identified by their retention time, absorption spectrum and molar extinction coefficients as described earlier [38].

### 2.6. RNA Extraction and RT—PCR

RNA isolation was performed by the TRIzol method (Sigma, USA) according to the manufacturer’s recommendations. The quantity and quality of the total RNA were determined using a NanoDrop 2000 spectrophotometer (Thermo Fisher Scientific, USA). cDNA synthesis was performed using the M–MLV Reverse Transcriptase Kit (Fermentas, Waltham, MA, USA), the oligo (dT) 21 primer and the random-6 primer. The expression patterns of the genes were assessed using the CFX96 Touch™ Real–Time PCR Detection System (Bio-Rad, Hercules, CA, USA). The gene-specific primers used are listed in Appendix A. *POR1*—protochlorophyllide oxidoreductase 1; *PSY1*—phytoene synthase 1; *PAL1*—phenylalanine ammonia-lyase 1; *CHS*—chalcone synthase; *ANS*—anthocyanidin synthase; *HY5*—elongated hypocotyl 5; *rbcL*—ribulose-1,5-bisphosphate carboxylase/oxygenase large subunit (RuBisCO large subunit); *Elip—*early light-induced protein; *psbA*—photosystem II protein D1; *psbD*—photosystem II protein D2; *psbB*—photosystem II CP47 chlorophyll-binding protein; *psbC—*photosystem II CP43 chlorophyll-binding protein; *psbO*—photosystem II manganese-stabilizing protein; *psbP*—oxygen evolving enhancer 2 of photosystem II; *psbQ*—photosystem II oxygen-evolving complex protein 3; *CAB1* PSII—light-harvesting complex of photosystem II were selected using nucleotide sequences from the National Center for Biotechnology Information (NCBI) database (www.ncbi.nlm.nih.gov, accessed on 2 February 2023, Bethesda, MD, USA) and phtyotozome.org with Vector NTI Suite 9 software (Invitrogen, Carlsbad, CA, USA). The gene transcript levels in needles and cambium were normalized to the expression of the *Tubulin1* gene. The experiments were performed with six biological and three analytical replicates.

### 2.7. Statistical Data Processing

The experiments included 3–5 biological replicates and 5–7 analytical replicates. The significance of the differences among the experimental groups was calculated by one-way analysis of variance (ANOVA) followed by Duncan’s method using SigmaPlot 12.3 (Systat Software Inc., Chicago, IL, USA). The letters above the columns indicate significant differences among the four different options at specific light intensities and times (0, 24, 48, and 72 h) (*p* < 0.05). The data are shown as the mean ± SD.

## 3. Results

### 3.1. Plant Phenotype

During the experiment, *hp* and WT plants accumulated more anthocyanins, and on the third day, the new leaves of the *hp* mutant had an intense purple color. The *lp* mutant exhibited miniaturization of leaf blades and yellowing of leaves caused by the destruction of chlorophyll. The *cry1* mutant showed no visible changes in leaf morphology.

### 3.2. Photosynthesis and Transpiration Rates, Stomatal Conductance and Chlorophyll Fluorescence Parameters

Initially_,_ the rates of photosynthesis (P_n_) were approximately the same and ranged from 3–5 µmol CO_2_ m^−2^s^−1^ (Table 1).

After 24 h of BL illumination at 500 μmol (photons) m^−2^s^−1^, the highest P_n_ value among the mutants was found for the *hp* mutant, and the lowest among all the variants after 72 h was found for the *cry1* mutant. At a higher light intensity, the photosynthesis rate value in the *cry1* mutant was the smallest after 72 h, while the P_n_ value of the *lp* mutant was the highest among all the variants after 48 and 72 h of growth under BL (Table 1).

It should be noted that photosynthesis was greater in the *hp* mutant than in the *lp* mutant only at the lowest dose of BL (500 μmol (photons) m^−2^s^−1^, 24 h) and that in all variants, except for *cry1*, photosynthesis increased as the plants were kept on BL.

Initially, the E values were higher for *lp* than for *hp* and *cry1* and were in the range of 1.0–1.9 mmol/m^2^s (Table 1). After 24, 48, and 72 h of BL illumination at 500 μmol (photons) m^−2^s^−1,^ the lowest E value among the variants was found in the *cry 1* mutant, and after 72 h the highest E value among all variants was observed in the WT and *lp* mutant. Additionally, E value in the *cry1* mutant was the smallest after 48 and 72 h expose to higher light intensity (1000 μmol (photons) m^−2^s^−1^), while in the *lp* mutant, it was the highest after 48 and 72 h of cultivation under BL.

Initially, the values of Gs were higher for *lp* than for *hp* and *cry1* and were in the range of 44–82 mmol m^−2^s^−1^ (Table 1). After 24, 48 and 72 h of BL illumination at 500 μmol (photons) m^−2^s^−1^, the lowest value of Gs among variants was found in the *cry1* mutant, and the highest after 72 h among all variants in the WT. At 1000 μmol (photons) m^−2^s^−1^, the Gs value in the *cry1* mutant was the smallest after 24, 48, and 72 h, while in the *lp* mutant, it was highest after 48 and 72 h (Table 1).

Light saturation curves for P_n_ showed that under the influence of BL, the P_n_ values at all points were the lowest for the *cry1* mutant and the highest for the *lp* mutant (Appendix A).

### 3.3. PAM Parameters

Initially, the values of the effective quantum yield Y(II) differed little among the WT, *cry1*, *hp*, and *lp* mutants (Figure 2). After 72 h of exposure to 500 and 1000 μmol (photons) m^−2^s^−1^, the Y(II) value was the lowest in the *cry1* mutant; in the other variants, it differed little.

The magnitude of NPQ was higher in the *lp* mutant than in the other mutants and the WT. The NPQ value was the highest in the *cry1* mutant after 72 h of expose at 1000 μmol (photons) m^−2^s^−1^, while the NPQ values in the other variants were approximately the same.

Before blue light (BL) irradiation, the wild type (WT) and *lp* mutant plants exhibited the highest values of Y(NO) and Y(NPQ), respectively. After 72 h of irradiation at 500 or 1000 μmol (photons) m^−2^s^−1^, the *cry1* mutant showed the highest Y(NO) and Y(NPQ) values, except for Y(NO) at 1000 μmol (photons) m^−2^s^−1^, where the differences in Y(NO) values among the WT and mutants were not significant. After 72 h of irradiation at 500 and 1000 μmol (photons) m^−2^s^−1^, the F_v_/F_m_ value was the smallest in the *cry1* mutant; in the other variants, no difference was found (Figure 3).

### 3.4. Pigment Contents

Initially, the Chl *a* content was greater in the WT and *hp* mutant than in the *cry1* and *lp* mutants (Table 2). Additionally, the *cry1* mutant had the lowest content of Chl *b* and carotenoids. After 24 h at 500 μmol (photons) m^−2^s^−1^ Chl *a*, *lp* reached its highest value and the content was the lowest in WT and *hp* mutant. The difference in the content of Chl *b* and carotenoids was not as noticeable. However, after 72 h, the Chl *a* content was almost 1.5 times higher in WT than in *cry1* and *hp*.

After 48 h under 1000 μmol (photons) m^−2^s^−1^ light, the contents of Chl *a* and Chl *b* were the highest in *cry1* and the lowest in *hp*, which were 1.4 and 1.7 times lower for Chl *a* and Chl *b*, respectively. At the same time, the carotenoid content was also the highest in *cry1* compared to the other variants. After 72 h, the content of Chl *a* in *hp* and *cry1* was lower than that in WT and *lp*, and that of Chl *b* was 1.8 times lower than in other variants. Additionally, the carotenoid content in *hp* was the lowest.

Initially, the phenolic content was greater in the WT and *hp* mutant than in the *cry1* and *lp* mutants. After 48 and 72 h at 500 μmol (photons) m^−2^s^−1^, the content of phenols was the lowest in the *cry1* mutant and the highest in the *hp* mutant. After 24 h, the content of phenols was the lowest in the WT, and after 24 h, it was the lowest in the WT and *cry1* mutant. After 72 h, the lowest content of phenols was found in the *cry1* mutant, and the highest in the *hp* mutant. A similar trend for *hp* and *cry1* mutants was detected by TEAC activity (Figure 4).

Thus, after 48 and 24 h, the *hp* mutant exhibited the highest activity, and the *cry1* mutant exhibited the lowest activity. At 500 and 1000 μmol m^−2^s^−1^ and 72 h of BL exposure, the highest anthocyanin content was detected in the *hp* mutant, and the lowest content was detected in the *cry1* mutant (Figure 4).

### 3.5. Carotenoid Composition

Lutein predominated in the WT and *hp* and *lp* mutants, and β-carotene predominated in *cry1* (Appendix A). Additionally, *cry1* had a high content of β-carotene and zeaxanthin after 72 h of exposure to BL (1000 μmol (photons) m^−2^s^−1^). A high zeaxanthin content was also detected in the *hp* mutant. The highest contents of neoxanthin and violoxanthin under these conditions were detected in the *lp* mutant.

### 3.6. Gene Expression

When analyzing genes, special attention was given to studying changes in the gene expression of those variants that showed the highest and lowest values of photosynthetic activity. After 24 h of exposure, the *hp* mutant was exposed to 500 or 1000 μmol (photons) m^−2^s^−1^. In the HILB group, the transcript levels of most of the studied genes increased significantly. This increase in expression was observed both for the genes encoding enzymes involved in the biosynthesis of secondary metabolites and photosynthetic pigments, including *POR1*, *PSY1*, *PAL1*, *CHS*, and *ANS*, as well as for the transcription factor *HY5* and the light-dependent protein *ELIP*, as well as for intrinsic and extrinsic photosystem II proteins such as *PSBA*, *PSBD*, *PSBC*, *PSBP*, *PSBQ*, and *rbcL* (Figure 5A,B).

After irradiation with 1000 μmol (photons) m^−2^s^−1^ for 24 h, the *hp* mutant tended to exhibit increased expression of these genes. The exceptions were the *psbO* and *psbB* genes, whose expression decreased in the *hp* mutant upon exposure to 500 or 1000 μmol (photons) m^−2^s^−1^ for 24 h.

According to the data obtained, in the *lp* mutant on the first day at 500 μmol (photons) m^−2^s^−1^ of the experiment, the expression of most genes hardly increased. After 24 h of irradiation at 1000 μmol (photons) m^−2^s^−1^, a further decrease in the activity of most genes was observed, but the expression of the *CHS* gene increased (Figure 5A,B).

By the third day of the experiment, under illumination with 500 μmol (photons) m^−2^s^−1^, the activity of most genes of the *hp* mutant either decreased or remained unchanged, excluding the *POR1* gene, where the expression significantly increased. At 1000 μmol (photons) m^−2^s^−1^, the transcript levels of almost all the genes decreased or remained unchanged.

After 72 h in the *lp* mutant, under 500 μmol (photons) m^−2^s^−1^ of light, the activity of the *POR1* gene increased, and under 1000 μmol (photons) m^−2^s^−1^, the activity of the *CAB1*, *POR1*, *psbQ*, *psbO*, *psbP*, and *psbB* genes increased compared with the other variants. At the same time, in the *cry1* mutant, the expression levels of most genes, particularly those in the 1000 μmol (photons) m^−2^s^−1^ variant after 72 h, were lower than those in the WT (Figure 5A,B).

## 4. Discussion

The *hp* mutants are known to contain increased levels of carotenoids and flavonoids [31]. Our studies have shown that the *hp* mutant grown under fluorescent lamps has increased levels of these pigments both before and after growth under HIBL. The *lp* mutant exhibited reduced levels of photosynthetic pigments, phenols, and anthocyanins (Table 2). A study of the effect of high-intensity white light (HIL) on photosynthetic processes in tomato plants showed that HIL (24 and 72 h, 1000 μmol (photons) m^−2^s^−1^) leads to an increase in the rate of photosynthesis but also to a decrease in PSII activity (indicators F_v/_F_m,_ PI_ABS_ and Y(II)), as well as to a decrease in the content of photosynthetic pigments in the hp mutant and to their increase in the *lp* mutant [26]. In addition, PSII resistance to HIL and increased photosynthetic rate were more pronounced at *hp*, likely due to the increased content of pigments such as carotenoids and flavonoids, including anthocyanins, which contribute to high adaptive potential. It is assumed that resistance and adaptation to HIL depend not only on the content of pigments but also on light irradiation conditions, which affect the proportion of the active form of CRY1 in its total pool. To do this, we compared the effects of blue HIL, in which the active CRY1 content is maximally high, using *hp* and *lp* mutants and a CRY1-deficient mutant.

The PSII activity of plants grown under fluorescent lamps and exposed to HIBL at intensities of 500 and 1000 μmol (photons) m^−2^s^−1^ decreased only in the *cry1* mutant, whereas in other cases, the changes were less significant. Apparently, the *cry1* mutant exhibits a dynamic type of photoinhibition, which is characterized by a loss of maximum photosynthetic efficiency due to changes in PA activity that are reversible and associated, in particular, with changes in the magnitude of NPQ [39]. Indeed, the NPQ value after exposure to BL was highest for the *cry1* mutant (0.9–1.0) (Figure 2). The NPQ mechanism is a protective mechanism that reduces photosynthetic electron transport by increasing the dissipation of absorbed light energy into heat, which reduces the generation of ROS in the photosynthetic electron transport chain. At the same time, in the mutant under BL, there was a significant reduction in PSII activity at the highest BL dosage (72 h at 1000 μmol (photons) m^−2^s^−1^). This increase was accompanied by an increase in the Y(NPQ) value, indicating enhanced light-induced, zeaxanthin-dependent thermal energy dissipation. Xanthophyll zeaxanthin is known to play a key role in protecting PA from oxidative damage by dissipating excess light energy [18], and its accumulation in *hp* mutants may be one of the protective mechanisms against oxidative stress. However, exposure to BL in the other variants did not result in a marked decrease in PSII activity, as in the CRY-deficient mutant. Apparently, the protective role in this case is played by the increased content of phenols, anthocyanins, and carotenoids, which can serve as both protective filters, absorbing excess light, and acting as antioxidants in *lp* and *hp* mutants, especially in *hp* compared to *cry1* mutants [19,40]. In addition to the genes encoding enzymes for the biosynthesis of pigments, the increased expression of the *HY5* transcription factor gene may also play a role in increasing the content of phenols and anthocyanins in hp, and high expression of the photoprotective gene may also play a role in maintaining the photosynthesis protein *Elip*, which is the most active in this mutant under these conditions (Figure 5A).

For specific carotenoids, the hp mutant actively synthesizes lutein, alpha- and beta-carotene, as well as zeaxanthin, which was further confirmed by an increase in the expression of the *PSY1* gene associated with carotenoid biosynthesis on the first day of the experiment. In the case of the lp mutant, neoxanthin likely plays a special role in the protection of photosystem 2 (Appendix A). Neoxanthin has been shown to be particularly effective in scavenging superoxides generated in the Mehler reaction, the rate of which is known to increase under abiotic stress conditions [23,41].

Extrinsic proteins, such as psbO, psbQ, and psbP, are associated with the luminal side of PSII and play key roles in the maintenance of clusters of oxygen-evolving complexes, the structural integrity of PSII and its optimal function, and the regulation of the PSII repair cycle [42]. Intrinsic proteins such as psbD, psbC, psbA, and psbB also contribute to the protection of PSII from light stress. After 24 h of irradiation with intense blue light (500 μmol (photons) m^−2^s^−1^), the *hp* mutant presented the highest expression levels of the *psbD* and *psbC* genes. It is assumed that an increase in the amount of these proteins may serve as one of the mechanisms that maintains high photosynthetic activity under bright light conditions. This concept is supported by the high levels of expression of the *psbD* and *psbC* genes in the *hp* mutant (Figure 5B).

Significant changes in the rate of photosynthesis, as well as in the stomatal conductance (Gs) and transpiration rate (E), were more pronounced than those in photochemical processes when plants were exposed to BL (Table 1). Among all the variants, after growing under BL, the lowest photosynthetic rate was found in the *cry1* mutant, while at a high dose of BL (48 and 72 h, 1000 μmol (photons) m^−2^s^−1^), the highest photosynthetic rate (P_n_) was observed in the *lp* mutant (Table 1). In addition, the rate of photosynthesis increased with incubation under blue light. Similar trends were observed for stomatal conductance and transpiration rate, where maximum Gs and E values were recorded in the *lp* mutant at 1000 μmol (photons) m^−2^s^−1^ at 48 and 72 h. A correlation was also observed between the photosynthetic rate and stomatal conductance at moderate light intensity (500 μmol (photons) m^−2^s^−1^). The photosynthesis (P_n_) rate in the WT, *cry1*, and *hp* mutant plants decreased in the following sequence: WT > *hp* > *cry1*. A similar trend was recorded for the Gs and E values (Table 1 and Appendix A). Apparently, high stomatal conductance in the *lp* mutant is associated with a possible decrease in sensitivity to the action of abscisic acid (ABA). The typical effect of ABA in leaves is decreasing water loss by transpiration due to closing stomata [43]. Blue light is also known to induce stomatal opening under the influence of CRY and PHOT [44]. In the *lp* mutant, stomatal opening occurred under the influence of strong BL; however, stomatal closure was not observed, probably due to ABA deficiency.

Under the same conditions, however, when plants were exposed to white HIL, the photosynthesis rate was greater in the *hp* mutant than in the *lp* mutant [26], which is likely linked to a decrease in the fraction of BL in the light spectrum. It is worth noting that at a lower BL intensity (500 μmol (photons) m^−2^s^−1^), no increase in P_n_ was observed in the *lp* mutant (Table 1 and Appendix A). Thus, a significant increase in G_s_ and P_n_ occurs at a high dose of BL. We assumed that the increase in P_n_ and Gs under the influence of BL was associated with CRY1, since with BL deficiency, neither an increase in P_n_ nor an increase in P_n_ was detected (Table 1 and Appendix A).

Phenolic compounds, including anthocyanins, play a key role in protecting plants from various stresses, especially UV-B radiation and high light [45,46]. Anthocyanins, which absorb visible light and have an antioxidant function, serve as protective filters against excess radiation [40]. We assume that the increased content of anthocyanins and phenols in the leaves of *hp* mutants contributes to the maintenance of primary photochemical processes compared to that of CRY1, which indicates the high adaptive ability of these mutants (Table 2). Apart from the *cry1* mutant, we did not find a decrease in PSII activity in the other variants. This may be due to reduced adaptive potential when plants are grown with *cry1* deficiency. Fantini et al., 2019 highlighted the key role of tomato cryptochromes in influencing seed mass and the ability to regulate early seedling growth, including hypocotyl elongation and root development, in young plants [28]. A study by Ashikhmin et al. (2023) confirmed the important role of pigment content in protecting photosynthetic activity from high-intensity white light. Our study showed that CRY1 plays a major role in protecting PA from HIBL, and that the leaf pigment content also influences PA protection; this effect is likely dependent on light intensity and the proportion of blue light in the spectrum.

## 5. Conclusions

In this work, we observed two different strategies for the resistance of the photosynthetic apparatus to high-intensity blue light. The *hp* mutants at the beginning of irradiation at 500 and 1000 μmol (photons) m^−2^s^−1^ were able to activate the expression of a network of associated genes and transcription factors, which led to an increase in flavonoids and anthocyanins and a change in the qualitative composition of carotenoids, mainly lutein, zeaxanthin, alpha a-carotene, and beta-carotene, as well as to a significant, more than 6-fold increase in the activity of low-molecular-weight antioxidants (TEAC) compared to baseline. However, when plants are exposed to blue light for a longer period, this adaptation strategy is less advantageous than that of the *lp* mutant. The *lp* mutant showed a lower adaptive capacity under short-term exposure to BL than did the *hp* mutant, but there was increased resistance under HIBL (1000 μmol (photons) m^−2^s^−1^) and long-term exposure (48 and 72 h), accompanied by an increase in the expression of the *psbO* and *psbP* extrinsic genes, as well as the maximum accumulation of neoxanthin, which serves as a mechanism of adaptation to stress, and increased stomatal conductance led to an increase in the activity of the photosynthetic apparatus and maintenance of the intensity of photosynthesis under long-term BL exposure. It can be assumed that both mechanisms are directly or indirectly related to the active form of CRY1 since both of these mechanisms are manifested in the *cry1* mutant.

The use of short-term exposure to BL is a convenient mechanism that increases the antioxidant potential of the resulting beneficial products. This can also serve as the basis for the production of agricultural products with specific biochemical characteristics and increased antioxidant activity.

## Figures and Tables

**Figure 1 antioxidants-13-00605-f001:**
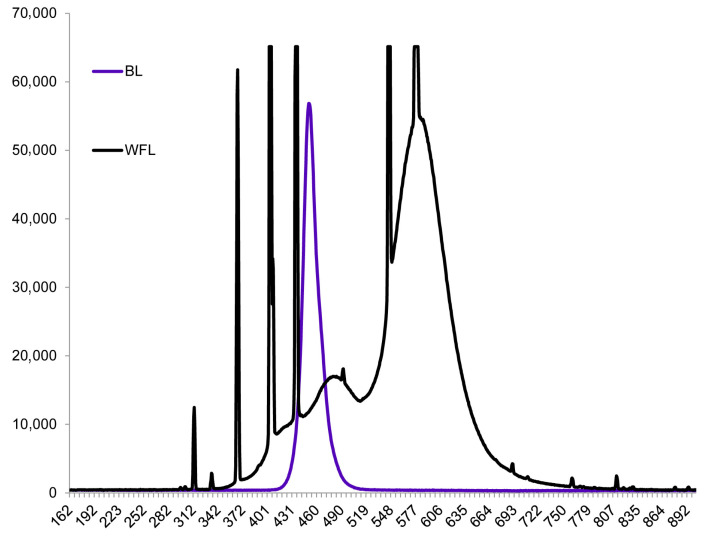
Spectra of the light sources used in the experiments.

**Figure 2 antioxidants-13-00605-f002:**
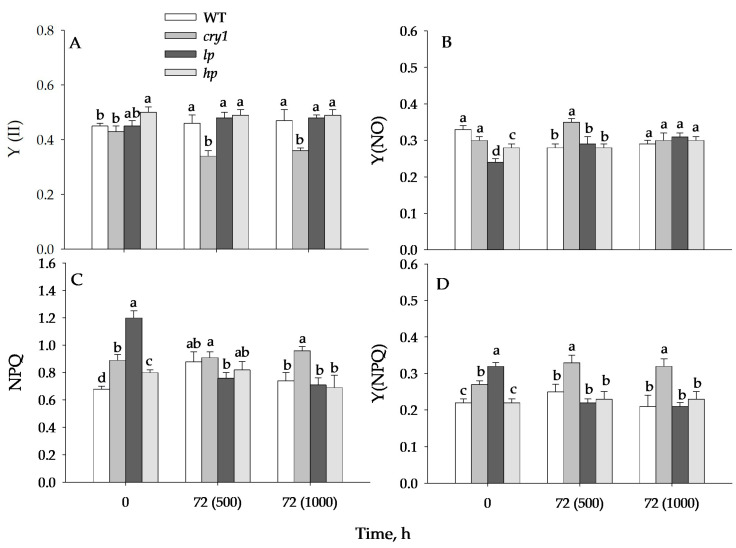
Impact of blue high irradiance exposure on fluorescence parameters: effective quantum PSII yield Y(II) (**A**), nonphotochemical quenching (NPQ) (**C**), Y(NO) (**B**) and Y(NPQ) (**D**) are PSII quantum yields of nonregulated and regulated nonphotochemical energy dissipation, respectively. Here. Y(NO) + Y(NPQ) + Y(II) = 1. The plants were grown under white fluorescence lamps for 42 days and then exposed to high blue light (I = 500 and 1000 μmol m^−2^s^−1^) for 72 h. The means ± SD are shown. Different letters within indicate significant differences (*p* ≤ 0.05) according to ANOVA on ranks followed by Duncan’s method within one particular variant of light and time, n = 6. WT = wild type; *cry1* = cryptochrome 1; *hp* = LA3005 high pigment mutant; *lp* = LA3617 low pigment mutant.

**Figure 3 antioxidants-13-00605-f003:**
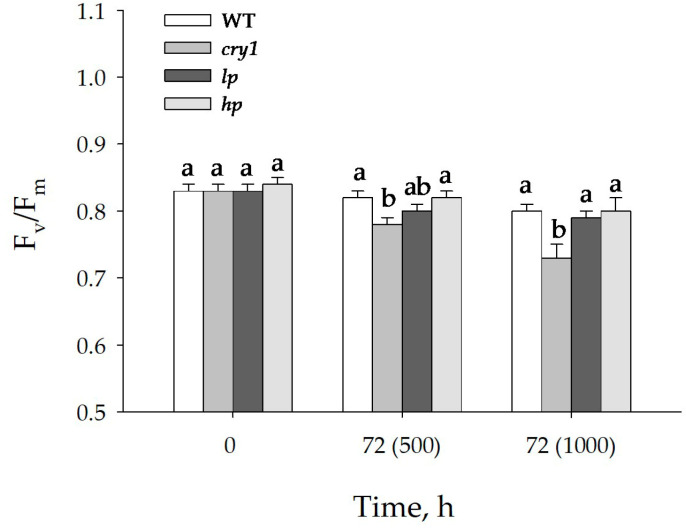
Changes in the PSII maximum quantum yield (F_v_/F_m_) of the WT and mutant plants to high intensity blue light. The plants were grown under white fluorescent light for 42 days and then exposed to high blue light (I = 500 or 1000 μmol m^−2^s^−1^) for 72 h. The means ± SD are shown. Different letters indicate significant differences (*p* ≤ 0.05) according to ANOVA on ranks followed by Duncan’s method within one particular variant of light and time, n = 6. WT = wild type; *cry1* = cryptochrome 1; *hp* = LA3005 high pigment mutant; *lp* = LA3617 low pigment mutant.

**Figure 4 antioxidants-13-00605-f004:**
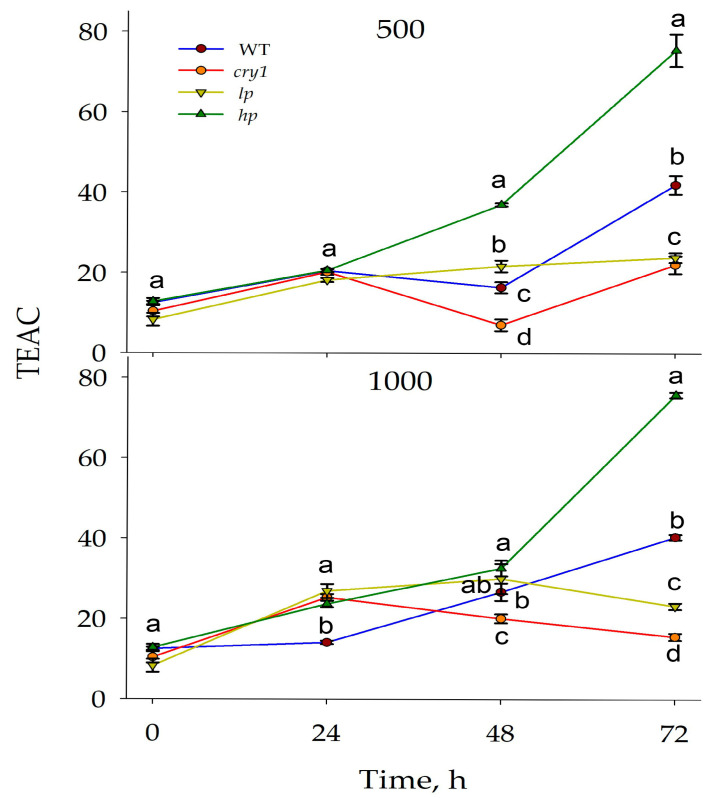
Changes in the Trolox equivalent antioxidant capacity (TEAC, µM Trolox g^−1^ FM) after the plants were exposed to high intensity blue light. The plants were grown under white lamps for 42 days and then exposed to high blue light (I = 500 or 1000 μmol (photons) m^−2^s^−1^) for 24, 48, and 72 h. The means ± SD are shown. Different letters within indicate significant differences (*p* ≤ 0.05) according to ANOVA on ranks followed by Duncan’s method within one particular variant of light and time, n = 3. WT = wild type; *cry1* = cryptochrome 1; *hp* = LA3005 high pigment mutant; *lp* = LA3617 low pigment mutant.

**Figure 5 antioxidants-13-00605-f005:**
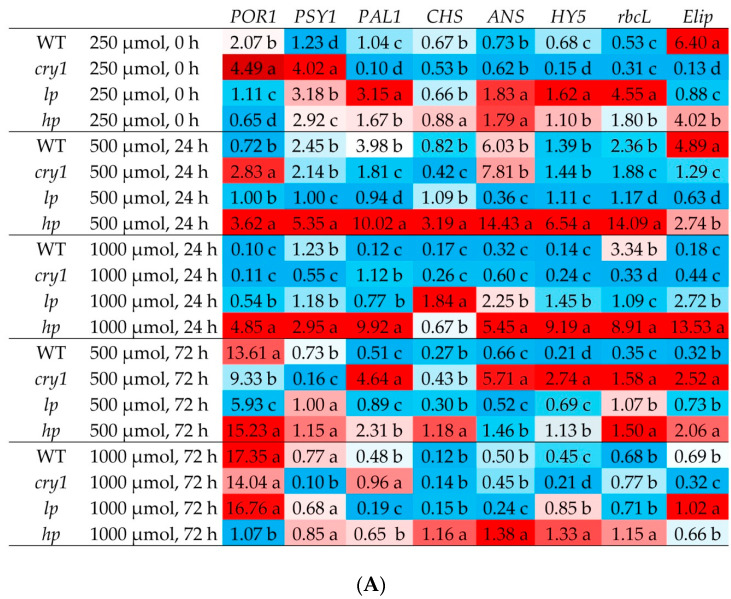
Transcript levels of protochlorophyllide oxidoreductase A (*PORA*), phytoene synthase (*PSY1*), phenylalanine ammonia-lyase (*PAL1*), chalcone synthase (*CHS*), anthocyanin synthase (*ANS*), elongated hypocotyl 5 (*HY5*), ribulose-1,5-bisphosphate carboxylase/oxygenase large subunit (*rbcL*), early light-induced protein (*Elip*) (**A**) and D1 protein of PSII (*psbA*), D2 protein of PSII (*psbD*), CP47 (*psbB*) and CP43 (*psbC*) proteins involved in the core antenna complex for light harvesting in PSII, the manganese-stabilizing protein essential for the stability of the water-splitting complex in PSII (*psbO*), proteins that optimize and regulate the oxygen-evolving complex of PSII (*psbP* and *psbQ*), and the chlorophyll a/b-binding protein of the light-harvesting complex in PSII (CAB1) (**B**). The plants were grown under white fluorescent light for 42 days and then exposed to high blue light (I = 500 or 1000 μmol m^−2^s^−1^) for 24 or 72 h. The means ± SDs are shown. Different letters within indicate significant differences (*p* ≤ 0.05) according to ANOVA on ranks followed by Duncan’s method within one particular variant of light and time, *n* = 3. WT = wild type; *cry1* = cryptochrome 1; *hp* = LA3005 high pigment mutant; *lp* = LA3617 low pigment mutant. The colors indicate the deviation from the mean expression level for each gene, represented by bars. A red color signifies an increase in expression more than twice the average. A white color denotes no significant change from the average. A blue color represents a decrease in expression more than twice the average.

**Table 1 antioxidants-13-00605-t001:** Impact of blue high irradiance exposure on photosynthesis (P_n_) and transpiration (E) rates and stomatal conductance (Gs) in WT and mutant plants. The plants were grown under white fluorescence lamps for 42 days and then exposed to high blue light (I = 500 or 1000 µmol m^−2^s^−1^) for 24, 48, and 72 h. The mean values ± SD are shown. Different letters indicate significant differences at *p* < 0.05, *n* = 6.

Option/Time (h)	0	24	48	72
P_n_ µmol CO_2_ m^−2^s^−1^				
WT (500)	4.4 ±0.9 a	5.6 ± 0.3 ab	9.2 ± 0.9 a	8.6 ± 0.5 a
*cry1* (500)	3.4 ± 0.2 a	3.4 ± 0.3 b	4.6 ± 0.9 b	3.1 ± 0.4 c
*lp* (500)	5.0 ± 0.7 a	4.3 ± 0.4 b	7.8 ± 0.5 a	7.0 ± 0.6 ab
*hp* (500)	3.8 ± 0.5 a	7.2 ± 0.5 a	6.7 ± 0.6 ab	6.2 ± 0.7 b
WT (1000)	4.4 ± 0.9 a	7.3 ± 0.7 b	8.8 ± 0.6 b	9.5 ± 0.6 b
*cry1* (1000)	3.4 ± 0.2 a	4.5 ± 0.7 c	3.4 ± 0.5 d	2.9 ± 0.3 d
*lp* (1000)	5.0 ± 0.7 a	10.1 ± 0.6 a	13.4 ± 0.8 a	11.9 ± 0.7 a
*hp* (1000)	3.8 ± 0.5 a	8.5 ± 0.5 ab	6.5 ± 0.5 c	5.9 ± 0.6 c
E mmol H_2_O m^−2^s^−1^				
WT (500)	1.35 ± 0.38 ab	1.33 ± 0.14 a	2.00 ± 0.28 ab	2.54 ± 0.31 a
*cry1* (500)	1.04 ± 0.07 b	0.83 ± 0.11 b	0.86 ± 0.09 c	1.22 ± 0.04 c
*lp* (500)	1.91 ± 0.28 a	1.42 ± 0.10 a	2.77 ± 0.31 a	2.08 ± 0.27 a
*hp* (500)	1.16 ± 0.10 b	1.44 ± 0.12 a	1.30 ± 0.15 b	1.44 ± 0.08 b
WT (1000)	1.35 ± 0.48 ab	1.58 ± 0.15 ab	1.90 ± 0.13 b	2.04 ± 0.11 b
*cry1* (1000)	1.04 ± 0.07 b	1.10 ± 0.16 b	0.97 ± 0.16 c	0.99 ± 0.09 d
*lp* (1000)	1.91 ± 0.38 a	2.13 ± 0.31 a	3.18 ± 0.29 a	2.94 ± 0.33 a
*hp* (1000)	1.16 ± 0.10 ab	1.79 ± 0.07 b	1.38 ± 0.15 bc	1.21 ± 0.07 c
G_s_ mmol m^−2^ s^−1^				
WT (500)	66.9 ± 8.9 ab	75.4 ± 7.1 a	123.1 ± 22.4 ab	164.6 ± 28.1 a
*cry1* (500)	44.3 ± 5.4 b	39.7 ± 7.2 b	44.9 ± 7.8 c	68.5 ± 3.3 c
*lp* (500)	82.3 ± 9.5 a	80.3 ± 5.4 a	134.3 ± 27.4 a	136.0 ± 14.1 a
*hp* (500)	67.1 ± 7.7 ab	79.8 ± 5.6 a	83.0 ± 12.5 b	89.6 ± 9.2 b
WT (1000)	66.9 ± 26.9 ab	80.2 ± 11.2 b	106.5 ± 6.3 b	117.3 ± 8.3 b
*cry1* (1000)	44.3 ± 5.4 b	54.0 ± 10.7 c	60.0 ± 9.6 c	43.8 ± 2.2 d
*lp* (1000)	82.3 ± 9.5 a	117.1 ± 10.3 a	187.9 ± 13.6 a	194.1 ± 30.1 a
*hp* (1000)	67.1 ± 7.7 ab	94.2 ± 7.2 ab	77.5 ± 6.9 c	67.0 ± 5.5 c

**Table 2 antioxidants-13-00605-t002:** The contents of the main photosynthetic pigments (Chl *a*, Chl *b* and Car mg g^−1^ DM) and anthocyanins (µg g^−1^ FM) in the leaves of tomato plants. The plants were grown under white lamps for 42 days and then exposed to high blue light (I = 500 or 1000 μmol m^−2^s^−1^) for 24 or 72 h. The mean values ± SD are shown. Different letters within indicate significant differences (*p* ≤ 0.05) according to ANOVA on ranks followed by Duncan’s method within one particular variant of light and time, *n* = 3.

		Chl *a*	Chl *b*	Car	Phenols	Anthocyanins
WT	250 µmol, 0 h	11.8 ± 0.6 a	3.9 ± 0.3 a	3.01 ± 0.14 ab	0.9 ± 0.1 a	1.1 ± 0.1 a
*cry1*	250 µmol, 0 h	8.3 ± 0.4 b	2.6 ± 0.2 c	2.25 ± 0.09 b	0.7 ± 0.1 b	0.9 ± 0.1 a
*lp*	250 µmol, 0 h	9.3 ± 0.7 b	3.3 ± 0.3 b	2.59 ± 0.31 b	0.7 ± 0.1 b	1.1 ± 0.1 a
*hp*	250 µmol, 0 h	12.8 ± 0.7 a	3.9 ± 0.4 a	3.24 ± 0.22 a	0.9 ± 0.1 a	1.3 ± 0.1 a
WT	500 µmol, 24 h	6.8 ± 0.3 b	2.3 ± 0.1 b	1.93 ± 0.15 c	1.5 ± 0.3 a	1.6 ± 0.1 b
*cry1*	500 µmol, 24 h	8.2 ± 0.5 a	3.3 ± 0.2 a	2.44 ± 0.07 b	1.4 ± 0.1 a	1.0 ± 0.2 c
*lp*	500 µmol, 24 h	9.5 ± 0.2 a	3.2 ± 0.1 a	2.66 ± 0.18 a	1.3 ± 0.1 a	0.9 ± 0.1 c
*hp*	500 µmol, 24 h	7.9 ± 0.4 b	2.3 ± 0.4 b	2.24 ± 0.16 c	1.5 ± 0.1 a	2.6 ± 0.4 a
WT	1000 µmol, 24 h	6.7 ± 0.5 ab	2.2 ± 0.3 b	2.89 ± 0.11 a	1.4 ± 0.5 b	1.5 ± 0.1 b
*cry1*	1000 µmol, 24 h	6.2 ± 0.5 b	2.1 ± 0.3 b	2.14 ± 0.17 c	1.8 ± 0.1 ab	0.8 ± 0.1 c
*lp*	1000 µmol, 24 h	7.6 ± 0.7 a	2.7 ± 0.4 a	2.42 ± 0.13 b	2.1 ± 0.2 a	1.4 ± 0.1 b
*hp*	1000 µmol, 24 h	7.1 ± 0.5 a	2.0 ± 0.3 b	2.01 ± 0.07 c	1.8 ± 0.2 ab	2.9 ± 0.2 a
WT	500 µmol, 72 h	10.8 ± 0.4 a	3.2 ± 0.2 a	2.07± 0.12 a	3.2 ± 0.1 a	2.1 ± 0.1 b
*cry1*	500 µmol, 72 h	7.2 ± 0.2 c	2.6 ± 0.1 b	1.82 ± 0.15 a	1.0 ± 0.3 c	0.8 ± 0.1 d
*lp*	500 µmol, 72 h	8.7 ± 0.2 b	2.5 ± 0.2 b	2.18 ± 0.22 a	1.5 ± 0.2 b	1.2 ± 0.1 c
*hp*	500 µmol, 72 h	7.3 ± 0.2 c	2.4 ± 0.2 b	2.11 ± 0.23 a	3.6 ± 0.2 a	3.4 ± 0.1 a
WT	1000 µmol, 72 h	8.2 ± 0.3 a	2.5 ± 0.2 a	2.42 ± 0.15 b	3.0 ± 0.1 b	2.9 ± 0.1 b
*cry1*	1000 µmol, 72 h	7.2 ± 0.2 b	2.2 ± 0.2 a	2.09 ± 0.14 c	0.9 ± 0.1 d	1.1 ± 0.1 d
*lp*	1000 µmol, 72 h	8.6 ± 0.2 a	2.5 ± 0.2 a	2.61 ± 0.13 a	1.8 ± 0.1 c	1.7 ± 0.1 c
*hp*	1000 µmol, 72 h	5.2 ± 0.2 c	1.4 ± 0.2 b	1.66 ± 0.13 d	4.1 ± 0.3 a	5.0 ± 0.3 a

WT = wild type; *cry1* = cryptochrome 1; *hp* = LA3005 high pigment mutant; *lp* = LA3617 low pigment mutant; Chl = chlorophyll; Car = carotenoids.

## Data Availability

Dataset available on request from the authors.

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
