# Peer review of "The Role of Pigments and Cryptochrome 1 in the Adaptation of Solanum lycopersicum Photosynthetic Apparatus to High-Intensity Blue Light"

_antioxidants, 2024, doi:10.3390/antiox13050605_

Round 1
Reviewer 1 Report
Long sentence in the beginning of abstract.
What is the significance of the current work? This should be added in the end of abstract and introduction.
I would prefer the content of Table 1 to be shown in the format of Figures. The same for Table 2.
Figure 4 The ANOVA should be added.
The overall conclusion of the key findings in the paper should concluded by making a conclusive figure. Currently, the discussion was made by discussing parameter one by one. However, what did the story tell us if we combine the results all together? This is more important than discussing the results of index one by one.
Long sentence in the beginning of abstract.
What is the significance of the current work? This should be added in the end of abstract and introduction.
I would prefer the content of Table 1 to be shown in the format of Figures. The same for Table 2.
Figure 4 The ANOVA should be added.
The overall conclusion of the key findings in the paper should concluded by making a conclusive figure. Currently, the discussion was made by discussing parameter one by one. However, what did the story tell us if we combine the results all together? This is more important than discussing the results of index one by one.
Author Response
We are grateful to the editor and reviewer for their attentive and objective attitude to the manuscript.
1. Long sentence in the beginning of abstract.
Answer: We reworked the abstract
2. What is the significance of the current work? This should be added in the end of abstract and introduction.
Answer: Necessary sentences were added to abstract and introduction
3. I would prefer the content of Table 1 to be shown in the format of Figures. The same for Table 2.
Answer: We fully concur with the reviewer that data visualization offers substantial benefits over tabular presentation. Unfortunately, in our case, incorporating both tables and figures would require an excessive amount of space. We appreciate the reviewer's understanding of these constraints.
4. Figure 4 The ANOVA should be added.
Answer: We are grateful to the Reviewer for identifying a significant oversight in our work. This error has now been corrected.
5. The overall conclusion of the key findings in the paper should concluded by making a conclusive figure. Currently, the discussion was made by discussing parameter one by one. However, what did the story tell us if we combine the results all together? This is more important than discussing the results of index one by one.
Answer: We fully agree with the Reviewer that the complexity and diversity of the data warrant a conclusive figure for clearer interpretation. Accordingly, we added a graphical abstract. We hope that this addition meets the reviewer's expectations.
Reviewer 2 Report
The submitted manuscript to be published in Antioxidants “ The role of pigments and cryptochrome in the adaptation of Solanum lycopersicum photosynthetic apparatus to high-intensity blue light” by authors Ashikhmin A et al represents an interesting and detailed investigation of the response of tomato plants Solanum lycopersicum in respect to photosynthetic activity to high light intensity blue light. The authors used wt of tomato and mutants with low and high carotenoid content. I find the approach very timing as the published data up to now concerning the pigment, especially of carotenoid content, are mainly focused on formation of fruits of agricultural plants, including tomato. In this respect the submitted manuscript is filling the gap with a focus on the impact of carotenoid content on photosynthetic performance.
The investigation was performed adequately, the manuscript is well structured, very well written and presented. The photosynthetic performance was followed, the quantity of pigments and phenolic metabolites were estimated as well as gene expression. The English language, style and gramma are extremely good.
I do not find a reason to add critical remarks.
I find the submitted manuscript worth publishing in the present form and it would represent a great interest for scientists interested in the role of increased and lower amounts of carotenoids under conditions of high light intensities and especially of blue light in respect to photosynthetic activity.
I didn’t find a reason to add critical remarks.
Author Response
We are grateful to the editor and reviewer for their attentive and objective attitude to the manuscript.
The submitted manuscript to be published in Antioxidants “ The role of pigments and cryptochrome in the adaptation of Solanum lycopersicum photosynthetic apparatus to high-intensity blue light” by authors Ashikhmin A et al represents an interesting and detailed investigation of the response of tomato plants Solanum lycopersicum in respect to photosynthetic activity to high light intensity blue light. The authors used wt of tomato and mutants with low and high carotenoid content. I find the approach very timing as the published data up to now concerning the pigment, especially of carotenoid content, are mainly focused on formation of fruits of agricultural plants, including tomato. In this respect the submitted manuscript is filling the gap with a focus on the impact of carotenoid content on photosynthetic performance.
The investigation was performed adequately, the manuscript is well structured, very well written and presented. The photosynthetic performance was followed, the quantity of pigments and phenolic metabolites were estimated as well as gene expression. The English language, style and gramma are extremely good.
I do not find a reason to add critical remarks.
I find the submitted manuscript worth publishing in the present form and it would represent a great interest for scientists interested in the role of increased and lower amounts of carotenoids under conditions of high light intensities and especially of blue light in respect to photosynthetic activity.
Answer: We are grateful to the reviewer for the careful consideration given to our manuscript and appreciate the time and kind words.